# Structural Fluctuations of the Human Proteasome α7 Homo-Tetradecamer Double Ring Imply the Proteasomal α-Ring Assembly Mechanism

**DOI:** 10.3390/ijms22094519

**Published:** 2021-04-26

**Authors:** Chihong Song, Tadashi Satoh, Taichiro Sekiguchi, Koichi Kato, Kazuyoshi Murata

**Affiliations:** 1Exploratory Research Center on Life and Living Systems (ExCELLS), National Institutes of Natural Sciences, 5-1 Higashiyama, Myodaiji, Okazaki, Aichi 444-8787, Japan; chsong@nips.ac.jp (C.S.); sekiguchi@ims.ac.jp (T.S.); 2National Institute for Physiological Sciences, National Institutes of Natural Sciences, 38 Nishigonaka, Myodaiji, Okazaki, Aichi 444-8585, Japan; 3School of Life Science, The Graduate University for Advanced Studies (SOKENDAI), Okazaki, Aichi 444-8787, Japan; 4Graduate School of Pharmaceutical Sciences, Nagoya City University, 3-1 Tanabe-dori, Mizuho-ku, Nagoya, Aichi 467-8603, Japan; tadashisatoh@phar.nagoya-cu.ac.jp; 5School of Physical Science, The Graduate University for Advanced Studies (SOKENDAI), Okazaki, Aichi 444-8787, Japan; 6Institute for Molecular Science, National Institutes of Natural Sciences, 5-1 Higashiyama, Myodaiji, Okazaki, Aichi 444-8787, Japan

**Keywords:** α7 subunit, cryo-electron microscopy, conformational fluctuations, double ring, 20S proteasome

## Abstract

The 20S proteasome, which is composed of layered α and β heptameric rings, is the core complex of the eukaryotic proteasome involved in proteolysis. The α7 subunit is a component of the α ring, and it self-assembles into a homo-tetradecamer consisting of two layers of α7 heptameric rings. However, the structure of the α7 double ring in solution has not been fully elucidated. We applied cryo-electron microscopy to delineate the structure of the α7 double ring in solution, revealing a structure different from the previously reported crystallographic model. The D7-symmetrical double ring was stacked with a 15° clockwise twist and a separation of 3 Å between the two rings. Two more conformations, dislocated and fully open, were also identified. Our observations suggest that the α7 double-ring structure fluctuates considerably in solution, allowing for the insertion of homologous α subunits, finally converting to the hetero-heptameric α rings in the 20S proteasome.

## 1. Introduction

Proteasomes are huge protein complexes involved in proteolysis in cells, and they are widely distributed across the three domains of life [1,2,3]. The 20S core particle of the proteasome preserves a common architecture composed of two heptameric α rings and two heptameric β rings, which are arranged into a four-layered αββα structure [4,5,6]. The proteolytic function of 20S core particle is executed within a cavity of this cylindrical chamber, which is dynamically controlled through opening the gate at the central pore of the α ring for substrate entry. In the closed form of the proteasome, the 20S core particle topologically blocks the entry of polypeptide substrates with the N-terminal tail of the α-subunits. To allow for substrate degradation, the core particle gate is opened upon association with proteasome activation factors [7,8,9,10,11,12].

In eukaryotic cells, this process is regulated by proteasome activating complexes, as best exemplified by the 19S regulatory particle. The 19S regulatory particles can attach at both ends of the 20S core particle, forming the 26S active proteasome, and thereby recognizing substrates and promoting their unfolding and translocation in an ATP-dependent manner [9,10,11].

Structural dynamics of the 20S core particle and its complex with the regulatory particles are characterized by cryo-electron microscopy (cryo-EM) [9,10,11,12], atomic force microscopy [13,14,15], and NMR spectroscopy [16,17], revealing the dynamics of the gate opening of the 20S core particle by interaction with activators.

In archaea, the α ring consists of seven identical α subunits, whereas the β ring is composed of one or two kinds of β subunits [6,17]. Archaeal proteasomal subunits are autonomously assembled into the 20S core particle [18]. In contrast, the α and β rings in eukaryotic proteasomes are hetero-heptamers, composed of seven different but homologous subunits, named α1–α7 and β1–β7 [5,9,10,11]. The assembly of these 28 subunits into the 20S core particle (seven α subunits in each of the two α rings and seven β subunits in each of the two β rings) does not proceed spontaneously; rather, it is assisted by multiple dedicated chaperones that function as molecular matchmakers and checkpoints [19,20].

Interestingly, the human α7 subunit, one of seven heterogeneous α subunits, self-assembles exclusively into a homo-tetradecamer with a double heptameric ring structure [21,22,23,24], while the α1–α6 subunits are under equilibrium between the monomeric and dimeric forms in vitro [25]. In the crystal structure of the α7 homo-tetradecamer, the two α rings are tightly connected with each other via hydrophobic and electrostatic interactions mediated by extended α-helices, which are involved in the interactions with the β subunits in the 20S core particle [22]. Similar to the interdigitation between the α-ring and β-ring, the crystal structure of the two α rings are structurally stabilized. On the other hand, the results of a previous high-speed atomic force microscopy analysis revealed that stacked α7 double-ring structures are unstable and wobble significantly in solution [26]. Further, the mobile α7 double ring can be disrupted by the addition of α4 and α6 subunits [22,25]. The biological significance of the self-assembling property of α7 remains elusive. However, these findings imply that the α7 double ring fluctuated in solution has a special mechanism for preventing and/or disassembling the formation of α7 homo-oligomers and converting them to the hetero-heptameric α-rings.

In this study, we performed cryo-EM analysis to delineate the structural fluctuations of the human proteasome α7 homo-tetradecamer in solution. The results provide mechanistic insight into the molecular interactions between the proteasome components that are important for the assembly of the functional 20S proteasome.

## 2. Results

### 2.1. The α7 Tetradecamer Double Ring Shows Three Different Structures in Solution

The structure of the α7 tetradecamer in solution was investigated by using single particle cryo-EM. Approximately 80,000 particles picked from 100 micrographs were subjected to 2D classification (Appendix A). The 2D class averages showed a dislocated α7 double ring called “dislocated” and a symmetric one called “symmetric” (Figure 1A). A fully open α7 double ring called “open” was also identified. The ratios of these three distinct types in 2D images were 23% symmetric, 62% dislocated, and 5% open (Figure 1B). “Others” includes intermediate particles between each type, broken particles, or unknown particles ((d) in Figure 1B). In these particle images, no single-ring structure was apparently observed. Based on the 2D class average images and the crystal structure of the α7 tetradecamer (PDB ID: 5DSV), the 3D structures of the symmetric, dislocated, and open types were reconstituted at resolutions of 5.9 Å, 8.1Å, and 12.1 Å, respectively (Figure 1C and Appendix A). All three cryo-EM maps confirmed that each α7 ring consisted of seven subunits, although the two rings were differently connected (Figure 1C). The symmetric type showed that the D7-symmetric double ring was approximately similar to the previously reported crystal structure. Regarding the dislocated type, the two heptameric rings were offset by 11 Å. Regarding the open type, the two rings were fully opened, appearing like a cooked clam, with an open angle of approximately 40°.

### 2.2. The α7 Subunit Models Were Fitted to the Densities in Each Cryo-EM Map

The crystal structure of the α7 tetradecamer shows a D7-symmetric double ring, and each subunit is composed of six α-helices and two β-sheets, where a pair of three α-helices sandwiches two β-sheets [22]. The density of the single α7 subunit was segmented from the cryo-EM map, and the crystallographic model (PDB ID: 5DSV) of the α7 subunit was fitted to the map (Figure 2A). All α-helices and β-sheets of the crystallographic model showed close agreement with the map with a few exceptions (Figure 2A,B; Appendix A). Fourteen copies of the refined model of the α7 subunit were individually fitted to the densities in each 3D cryo-EM structure of the symmetric, dislocated, and open types of the α7 tetradecamer double ring (Figure 2C–E). Although the resolutions of the dislocated and open cryo-EM maps were limited compared to that of the symmetric cryo-EM map, the α-helix densities were easily identified, showing the following cross-correlation values with the fitted models; 0.86 for the symmetric type, 0.81 for the dislocated type, and 0.73 for the open type.

### 2.3. Cryo-EM Structures of the α7 Double Ring in Solution

Interestingly, the cryo-EM map of the symmetric type double ring was approximately similar to the crystallographic model, but the interaction points between the two rings exhibited a large difference between the two structures (Figure 3). Compared to the crystal structure, the symmetric type double ring of the cryo-EM map was stacked with a twist of approximately 15° and a separation of approximately 3 Å between the two rings (Figure 3A,B). In addition to this deformation, the fitted model of the dislocated type showed that the two rings were offset by approximately 11 Å (Figure 3C,D). Regarding the open type, the two rings of the dislocated type were further opened by approximately 40° via the interaction of the two α7 subunits at one end of each ring (Figure 2E). The results suggest that the twisting and separation of the α7 double ring in solution cause further dislocated and open structures. In the crystallographic model, the two rings interacted strongly via a single α-helix located at the inner surface of each ring (Figure 3E). However, in the symmetric type, the interaction occurred in a smaller area of the two α-helices (Figure 3F). Furthermore, in the dislocated and open types, the two rings interacted at the tips of the α-helices (Figure 3G,H). Interactions between two α rings also occurred in a smaller area compared to the interaction of the α- and β- rings (Appendix A).

## 3. Discussion

In this study, we investigated the double-ring structure of the α7 homo-tetradecamer in solution using cryo-EM. The results showed that the structures were different from the previously reported X-ray crystallographic model [22]. Compared to the crystal structure, the symmetric-type cryo-EM map showed that the double ring twisted 15° and there was a separation of 3 Å between the two rings (Figure 3A,B). It was shown that a new molecular interaction was created between the α7 homo-heptameric rings, where the two rings interacted in a smaller area of the α-helices (two turns of the helices), compared to what was seen in the crystallographic model (three turns of the helices) (magenta lines in Figure 3E,F). The interaction areas were further shifted to the tips of the α-helices (one turn in the helices) in the dislocated and open types (Figure 3G,H). From these observations, we propose a structural fluctuation model of the α7 homo-tetradecamer in solution (Figure 4; Appendix A), where sequential conformational changes possibly occur via helix–helix interactions. Compared to the crystallographic model, the α7 homo-tetradecamer dominantly exhibits a twist of 15° and a separation of 3 Å between the two rings in solution, in which the helix–helix interactions between the two rings occurred within two helical turns. Regarding the dislocated type, one ring slid by reducing the helix–helix interaction area from two turns to one, resulting in an offset of 11 Å between the two rings. Regarding the open type, the two rings further opened while retaining the helix–helix interaction on one side of the ring, thereby exhibiting a cooked clam-like structure. In this structural change in the same solution, it is assumed that there is no change in the composition of the α-ring itself, only the positional relationship of the two rings changes. Like the crystal structure [22], the cryo-EM data revealed that α7 homo-tetradecamer has a central pore in the α7-ring (Figure 1 and Figure 2) as observed in the gate-open structures of 20S core particle complexed with proteasome activators [7,9,10,11]. The relative positions between the two α-rings could be altered regardless of the potential dynamics of the gate opening in the α-ring, which is caused by the conformational changes of the N-terminal tails. In the crystallographic model, the double-ring structure may artificially create the strong helix–helix interactions in the crystal packing. In the future, higher resolution of the α7 homo-tetradecamer in solution is necessary to clarify the exact molecular interactions between the two rings.

Previous biochemical and biophysical analysis results have demonstrated that the α7 homo-tetradecamer exclusively formed the double-ring structure and is disassembled into homo-heptameric single rings through associations with α4 or α6 subunits, giving rise to 1:7 hetero-octameric complexes of α4-α7 or α6-α7, respectively [22,25]. In addition, a previous high-speed atomic force microscopy study revealed that the disassembly consists of a two-step process [26]. Monomeric α6 initially cracks at the interface between the two stacked α7 single rings and is subsequently accommodated in the central pore of the α7 single ring. These findings demonstrate the versatile nature of the proteasomal α subunits with structural homology, giving deeper insights into the mechanisms behind assembly and disassembly of oligomeric proteins. The results of the present cryo-EM analysis together with the previous high-speed atomic force microscopy observations shed light on the conformational plasticity of the α7 homo-tetradecamer in solution. The intrinsic fluctuations of the α7 double-ring structure may facilitate its initial complex formation with α4 or α6, thereby contributing to its disassembly mechanism.

## 4. Materials and Methods

### 4.1. Sample Preparation for Cryo-EM

The human proteasome α7 subunit (PSMA3 [P25788]; residues 1–255) was expressed and purified as described previously [23,24]. The sample was dissolved in a buffer containing 20 mM Tris-HCl (pH 8.0) and 150 mM NaCl. An aliquot (2.5 μL) of sample solution was applied onto R1.2/1.3 MO 200 mesh holey grids (Quantifoil Micro Tools, Jena, Germany) coated with thin carbon membranes and pre-treated with glow discharge using a plasma ion bombarder (PIB-10, Vacuum Device, Mito, Japan) for 30 s. The grid was blotted for 4 s with a force level of 7 at 4 °C and 95% humidity, and then it was flash frozen in liquid ethane using a Vitrobot Mark IV system (Thermo Fischer Scientific, Hillsboro, OR, USA). The vitreous ice sample grid was maintained at liquid-nitrogen temperature within a JEM2200FS electron microscope (JEOL Ltd., Tokyo, Japan) using a side-entry Gatan 626 cryo-transfer holder (Gatan Inc., Pleasanton, CA, USA), and it was imaged using a field-emission gun operated at 200 kV and an in-column (Omega-type) energy filter operating in zero-energy-loss mode with a slit width of 20 eV. A total of 100 images were collected on a direct-detector CMOS camera (DE20, Direct Electron, LP, San Diego, CA, USA) at a nominal magnification of 40,000×, corresponding to 1.42 Å per pixel on the specimen.

### 4.2. Image Processing for Cryo-EM

The images were corrected for beam-induced motion with dose-weighting using the RELION 3.1 software [27], and their contrast transfer functions were estimated with the CTFFIND4 software [28]. From all of the images, 80,048 particles were collected using RELION 3.1 software. After 2D classification, the 2D averaged images were divided into the symmetric type (18,388), dislocated type (49,691), and open type (3842). The particles of the symmetric type were used to generate initial 3D models by imposing D7 symmetry. The best model was used as a reference for the following 3D refinement. The final 3D cryo-EM map of the symmetric type was reconstructed using the 18,388 particles for a resolution of 5.9 Å (gold standard Fourier shell correlation (GS-FSC)). For the dislocated type, the 2D classes of the dislocated type were subjected to 2D classification again, and the good classes, containing 23,703 particles, were used for 3D reconstruction and classification. Two good 3D classes, which included 16,558 particles, were ultimately selected. 3D refinement was carried out with these selected particles without imposing symmetry. For the open type, the 2D classes containing the open type images were used for 3D reconstruction and classification. A good class was selected, and the 3D map was refined using the particles from this class. The resolutions of the dislocated and open types were calculated to be 8.1 Å and 12.1 Å (GS-FSC), respectively. Volume renderings of the maps were created in UCSF Chimera [29]. For atomic model building, the map containing a subunit of the α7 homo-tetradecamer was extracted by using UCSF Chimera. The atomic structures of the individual α7 subunits in the double ring were manually re-built for each cryo-EM map based on the crystal structure using COOT [30], and the resultant models were refined by using PHENIX [31]. Data collection, image processing, and model statistics are summarized in Appendix A.

## Figures and Tables

**Figure 1 ijms-22-04519-f001:**
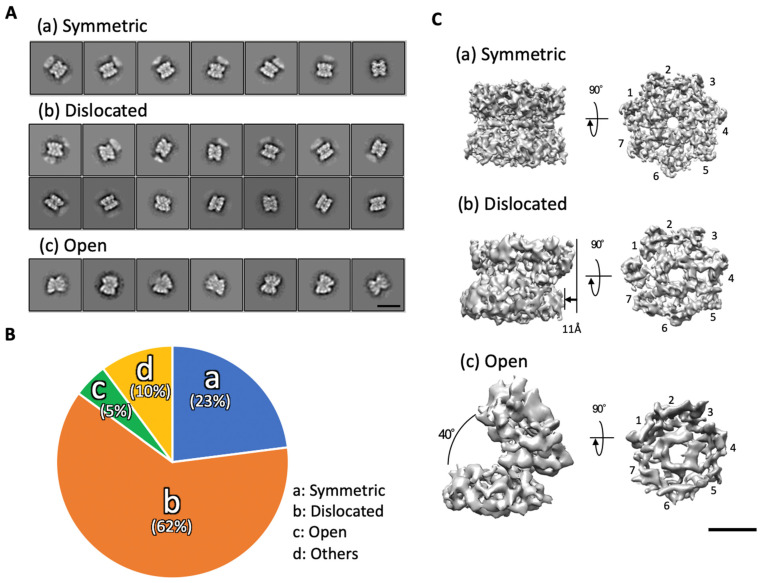
Single particle cryo-EM analysis of the α7 homo-tetradecamer. (**A**) Representative 2D class average images of the three types of the α7 homo-tetradecamer: (**a**) symmetric, (**b**) dislocated, and (**c**) open. Scale bar, 100 Å. (**B**) Ratio of each type of the α7 homo-tetradecamer classified by 2D classification (**a**–**c**). Others (**d**) include intermediate particles between each type, broken particles, or unknown particles. (**C**) 3D reconstructions of the three types of the α7 homo-tetradecamer. Scale bar, 50 Å.

**Figure 2 ijms-22-04519-f002:**
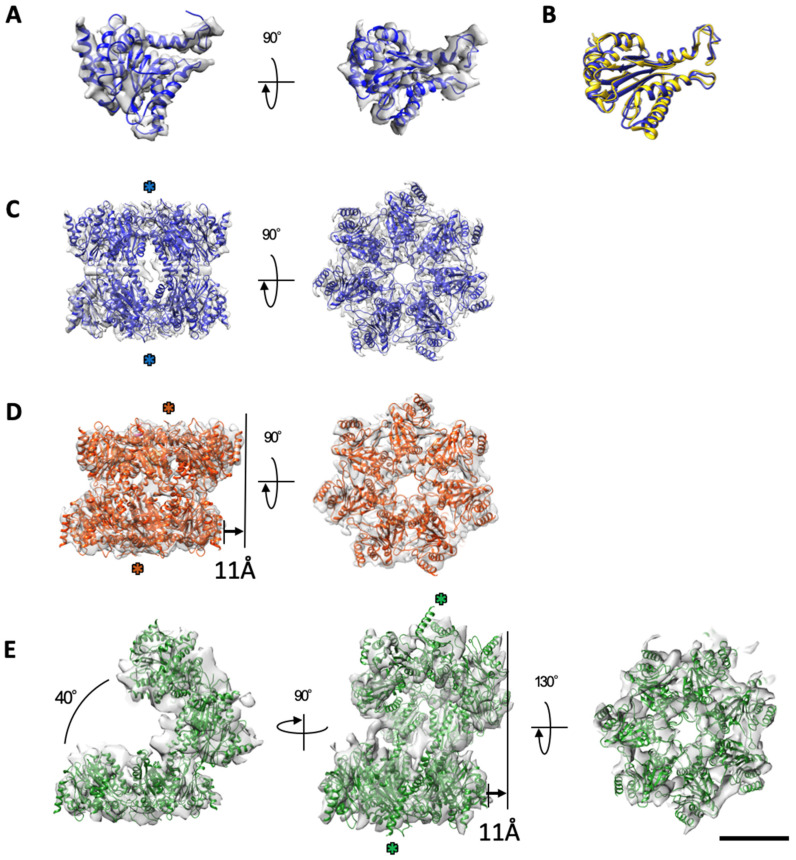
Assembled subunit conformation of each type of the α7 homo-tetradecamer. (**A**) The α7 homo-tetradecamer single subunit segmented from the symmetric-type cryo-EM map and the fitted model (blue) built from the X-ray crystal structure (PDB ID: 5DSV). (**B**) Comparison of the α7 subunit models by cryo-EM and X-ray crystallography. Ribbon diagrams of the subunits by cryo-EM and X-ray crystallography are colored by blue and yellow, respectively. (**C**–**E**) Assembled subunit conformations of each type of the α7 homo-tetradecamer: symmetric (blue in (**C**)), dislocated (red in (**D**)), and open (green in (**E**)). Scale bar, 50 Å. The double ring opens with a small twist (green asterisks).

**Figure 3 ijms-22-04519-f003:**
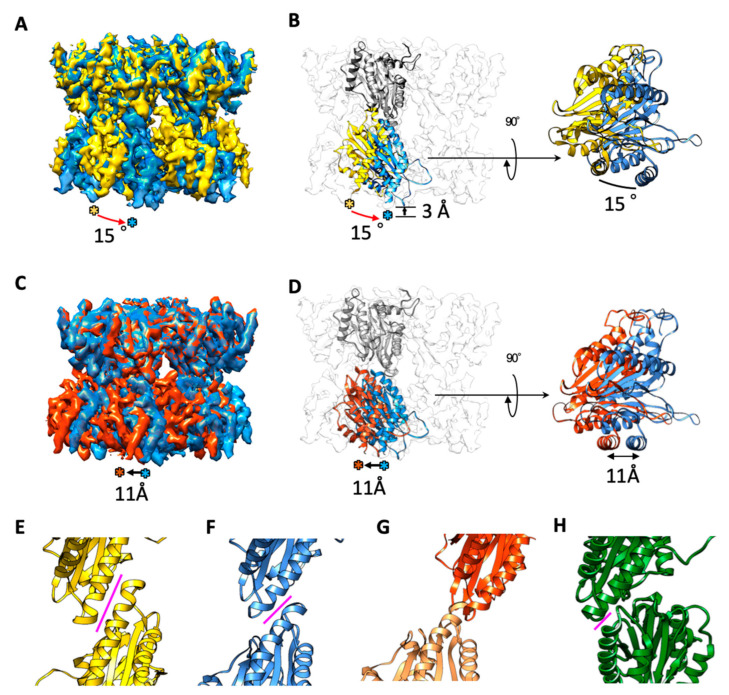
Structural comparison of the α7 homo-tetradecamer between the cryo-EM models and the X-ray crystallographic model. (**A**) Cryo-EM map of the symmetric-type α7 homo-tetradecamer (blue) was superimposed with the X-ray crystallographic model (PDB ID: 5DSV; yellow). When the upper ring of the double ring was overlapped at the same position, the dislocation of the lower ring was evaluated. (**B**) Ribbon diagrams of the α7 subunits in each lower ring are colored by blue (cryo-EM) and yellow (X-ray). The cryo-EM model shows a clockwise twist of approximately 15° and a separation of approximately 3 Å compared to the X-ray crystallographic model. (**C**) Cryo-EM map of the dislocated type of the α7 homo-tetradecamer (red) overlapped with that of the symmetric type (blue). When the upper ring of the double ring was overlapped at the same position, the dislocation of the lower ring was evaluated. (**D**) Ribbon diagrams of the α7 subunits in each lower ring are colored by blue (symmetric) and red (dislocated). The lower ring was offset by approximately 11 Å in the dislocated type. (**E**–**H**) Helix–helix interactions between the two rings of the α7 homo-tetradecamer: X-ray crystallographic model (**E**) and cryo-EM symmetric (**F**), dislocated (**G**), and open (**H**) types. The magenta line represents a possible interaction area. The length of the helix–helix interaction was measured as 13 Å in (**E**), 8 Å in (**F**), 9 Å in (**G**), and 6 Å in (**H**).

**Figure 4 ijms-22-04519-f004:**
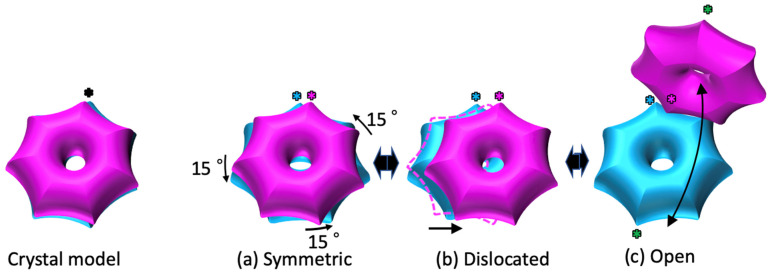
Structural fluctuation model of the α7 homo-tetradecamer in solution. The symmetric-type cryo-EM model (**a**) shows a twist of 15° Å and a separation of 3 Å between the two rings, compared to the crystallographic model (left panel). In the dislocated-type cryo-EM model (**b**), one ring further rotates and slides while retaining the interaction area on one side of the ring, resulting in an offset of 11 Å between the two rings. In the open-type cryo-EM model (**c**), the two rings open while retaining the interaction area on one side of the ring, resulting in a twisted, cooked clam-like structure. The helix–helix interaction areas are labeled by asterisks.

## Data Availability

Cryo-EM maps have been deposited in the Electron Microscopy Data Bank under accession numbers EMD-30990, EMD-30991 and EMD-30992. The atomic models have been deposited in the Protein Data Bank under accession number 7E55.

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
