# Peer review of "Structural Fluctuations of the Human Proteasome α7 Homo-Tetradecamer Double Ring Imply the Proteasomal α-Ring Assembly Mechanism"

_ijms, 2021, doi:10.3390/ijms22094519_

Round 1
Reviewer 1 Report
The work is an interesting continuation of earlier works from the group concerning the self-assembled single and double rings of alpha 7 subunit of the human 20S proteasome. According to the Authors the work suggests a mechanism for disassembly of the rings and insertion of other that alpha7 subunits for a potential formation of a native heteroheptameric ring. I have the following problems with data presentation:
- Overinterpretation of the "practical" significance of the work. There is no evidence that formation and disassembly of alpha7 homoheptamer occurs in vivo. I would tend to think that the machinery of specialized chaperones that assist in 20S assembly would prevent homoheptamer formation early enough. Also, to form the homoheptamer, an excess of single alpha7 subunits would need to be available - not necessarily a plausible scenario. To justify the significance, the Authors need to discuss it in much more detail.
- The Introduction and Discussion are very brief. There is a considerable body of literature about the structural dynamics of the 20S proteasome: NMR data, AFM studies (much more than the high-speed AFM imaging of the alpha7 rings), extensive cryoEM studies (26S but also attached 20S). The Authors need to show their data in the context of accumulated knowledge about proteasome dynamics. For example: does the enlargement of alpha7 ring opening during rings' twisting resemble gate opening in native 20S? Are contacts between the two alpha7 rings resemble contacts between alpha and beta rings in the native 20S? If there are any reasonable similarities, they should be presented in the Results section. If not, at least some discussion should be provided. By showing/discussing that, the work may gain a more universal significance.
- In Figure 1 examples of the "symmetric", "dislocated" and "open" types are shown. In the pie chart there is also a sizable "other" category (10%), not shown and not discussed. Are these particles intermediates between the other types? This should be specified.
Author Response
Reviewer1
The work is an interesting continuation of earlier works from the group concerning the self-assembled single and double rings of alpha 7 subunit of the human 20S proteasome. According to the Authors the work suggests a mechanism for disassembly of the rings and insertion of other that alpha7 subunits for a potential formation of a native heteroheptameric ring. I have the following problems with data presentation:
- Overinterpretation of the "practical" significance of the work. There is no evidence that formation and disassembly of alpha7 homoheptamer occurs in vivo. I would tend to think that the machinery of specialized chaperones that assist in 20S assembly would prevent homoheptamer formation early enough. Also, to form the homoheptamer, an excess of single alpha7 subunits would need to be available - not necessarily a plausible scenario. To justify the significance, the Authors need to discuss it in much more detail.
Authors’ response: Line 201: We concur with thank the reviewer. Although the biological significance of the self-assembling property of α7 remains to be elusive in the context of proteasome formation, our previous findings demonstrate versatile nature of the proteasomal α subunits with structural homology. We believe that the structural knowledge gained in this study will provide further insights into the molecular mechanisms behind assembly and disassembly of oligomeric proteins in general. In view of this, we revised the main text in the manuscript as follows.
" These findings demonstrate versatile nature of the proteasomal α subunits with structural homology, giving deeper insights into the mechanisms behind assembly and disas-sembly of oligomeric proteins."
- The Introduction and Discussion are very brief. There is a considerable body of literature about the structural dynamics of the 20S proteasome: NMR data, AFM studies (much more than the high-speed AFM imaging of the alpha7 rings), extensive cryoEM studies (26S but also attached 20S). The Authors need to show their data in the context of accumulated knowledge about proteasome dynamics. For example: does the enlargement of alpha7 ring opening during rings' twisting resemble gate opening in native 20S? Are contacts between the two alpha7 rings resemble contacts between alpha and beta rings in the native 20S? If there are any reasonable similarities, they should be presented in the Results section. If not, at least some discussion should be provided. By showing/discussing that, the work may gain a more universal significance.
Authors’ response: Thank you for your kind comments. We have added sentences in introduction, results, and discussion sessions. We also cited the literatures describing the structural dynamics of proteasomes of functional relevance in the revised manuscript.
The gate opening of 20S accompanies conformational change caused by the activator such as 19S regulatory particle. In this study, we reported structural fluctuations in the α7 homo-tetradecamer, which showed a change in the positional interactions between the two rings without a conformational change in the ring itself. Therefore, structural fluctuations of the α7 homo-tetradecamer in solution are not directly related to the structural change of the gate opening. We added this content in the Introduction and Discussion as follows.
Introduction: Line 37: " In the closed form of the proteasome, the 20S core particle topologically block the entry of polypeptide substrates with the N-terminal tail of the α-subunits. To allow for substrate degradation, the core particle gate is opened upon association with proteasome activation factors [7,8,9,10,11,12]. "
Discussion: Line 182: " In this structural change in the same solution, it is assumed that there is no change in the composition of the α-ring itself, only the positional relationship of the two rings changes. Like the crystal structure [22], the cryo-EM data revealed that α7 homo-tetradecamer has a central pore in the α7-ring (Figures 1, 2) as observed in the gate-open structures of 20S core particle complexed with proteasome activators [7,9,10,11]. The relative positions between the two α-rings could be altered regardless of the potential dynamics of the gate opening in the α-ring, which is caused by the conformational changes of the N-terminal tails."
In addition, a description related to the structure between the α and β rings was added, and a figure comparing the molecular interactions between the α-α ring and the α-β ring was newly included as Supplementary Figures S2.
Introduction: Line 64: "Similar to the interdigitation between α-ring and β-ring, the crystal structure of the two α rings are structurally stabilized."
Result: Line 127: " Interactions between two α rings also occurred in a smaller area compared to the inter-action of the α- and β- rings (Supplementary Figures S2)."
- In Figure 1 examples of the "symmetric", "dislocated" and "open" types are shown. In the pie chart there is also a sizable "other" category (10%), not shown and not discussed. Are these particles intermediates between the other types? This should be specified.
Authors’ response: Line 86: "Others" category includes intermediate particles between each type, broken particles, or unknown particles. We added it in the result section and the legend of Figure1B.
" "Others" includes intermediate particles between each type, broken particles, or un-known particles ((d) in Figure 1B). In these particle images, no single-ring structure was apparently observed."
Reviewer 2 Report
Song et al. reported the assembly mechanism of the human proteasome α7 homotetradecamer double ring using the Cryo-EM technique.
They clearly showed the structure of asymmetric-, dislocated- and open-state of α7 homotetradecamer. Although there is no detailed structural explanation at the atomic level due to its low resolution, it is very clear that the conformational change of proteasome α7 homotetradecamer provide new information compared to the existing crystallographic structures. In addition, the structural fluctuation model provided by the authors is judged to be a very reliable result based on Cryo-EM data. I believe the results of this study are very interesting to researchers and readers in related fields. Therefore, I recommend that the thesis be accepted after a few revisions.
1.Line 141: (Ishii et al. 2015) reference format needs to be modified.
2. The author should remove the red underline in Supplmentary TableS1.
3. I suggest that the author slow down the rate of change in the flucturation model of molecules in the Supplemenatry Movie. Meanwhile, I believe that adding text to each symmetric, dislocated and open state will help readers understand.
Author Response
Reviewer2
Song et al. reported the assembly mechanism of the human proteasome α7 homotetradecamer double ring using the Cryo-EM technique.
They clearly showed the structure of asymmetric-, dislocated- and open-state of α7 homotetradecamer. Although there is no detailed structural explanation at the atomic level due to its low resolution, it is very clear that the conformational change of proteasome α7 homotetradecamer provide new information compared to the existing crystallographic structures. In addition, the structural fluctuation model provided by the authors is judged to be a very reliable result based on Cryo-EM data. I believe the results of this study are very interesting to researchers and readers in related fields. Therefore, I recommend that the thesis be accepted after a few revisions.
Authors: We would like to thank the reviewer for their critical reading of the manuscript and their positive comments. We modified the manuscript according to the comments.
- Line 141: (Ishii et al. 2015) reference format needs to be modified.
Authors’ response: We have modified it to the reference number.
- The author should remove the red underline in Supplmentary TableS1.
Authors’ response: We have removed the red lines.
- I suggest that the author slow down the rate of change in the flucturation model of molecules in the Supplemenatry Movie. Meanwhile, I believe that adding text to each symmetric, dislocated and open state will help readers understand.
Authors’ response: Thank you for your suggestion. We have edited the Supplementary Video according to your comments. The movie have become much easier to understand.
Round 2
Reviewer 1 Report
The manuscript is fine; thank you very much for fixing.